# Targeting epigenetic regulators: In-silico discovery of natural inhibitors against histone demethylase KDM4C

Mukesh Kumar[1,2☯], Anusha P[3☯], Soumyadip Mukhopadhyay[4], Subarnarekha Chowdhury[1], Manoj Phalak[5], Uma Devi[4], Prakash K. Shukla[4]*

1 Department of Biophysics, All India Institute of Medical Sciences, New Delhi, India, 2 Department of Optometry, School of Allied Health Sciences, Galgotias University, Greater Noida, India, 3 School of Bio-sciences and Technology Vellore Institute of Technology, Vellore, Tamil Nadu, India, 4 Centre for Bio-Separation Technology, Vellore Institute of Technology, Vellore, Tamil Nadu, India, 5 Department of Neurosurgery, All India Institute of Medical Sciences, New Delhi, India

☯ The authors contributed equally to this work.
* prakash.shukla@vit.ac.in

## Abstract

Cancer is a multifaceted disease driven by genetic mutations and epigenetic dys-regulation. Among epigenetic modifiers, histone demethylases like KDM4C (lysine demethylase 4C) play a pivotal role in tumor progression by removing repressive methylation mark at Histone H3K9/H3K36 and altering chromatin structure and gene expression. Overexpression of KDM4C has been implicated in various malignancies, including breast, prostate, colorectal, and hepatocellular carcinomas, hence it is promising drug target. This study employs a structure-based drug discovery strategy to identify natural polyphenolic inhibitors of KDM4C. High-throughput virtual screening, followed by molecular docking, molecular dynamics (MD) simulations, and MM-GBSA free energy calculations, used to assess binding potential. Pectolinarin and compound 202 emerged as top candidates, outperforming the reference ligand (6X9) used from PDBID: 5KR7, in docking scores, and exhibiting robust hydrogen bonding and hydrophobic interactions within the active site. MD simulations over 200 ns confirmed complex stability, indicated by consistently low RMSD and RMSF values. MM-GBSA analysis revealed strong binding affinities with free energy values of −68.4 kcal/mol and −65.7 kcal/mol for Pectolinarin and compound 202, respectively. ADMET predictions supported their drug-likeness, suggesting favorable pharmacokinetic profiles, oral bioavailability, and low toxicity. These findings highlight pectolinarin and compound 202 as promising leads for KDM4C-targeted cancer therapy. Further experimental validation is required to confirm their efficacy and specificity. Overall, this work demonstrates the potential of computational approaches in advancing the discovery of nature-derived epigenetic therapeutics.

**Data availability statement:** Data are mentioned within the manuscript.

**Funding:** No funding issued for this work.

**Competing interests:** No Authors competing interest.

# 1. Introduction

Cancer remains a critical global public health challenge, claiming millions of lives each year and increasing in prevalence. As per the World Health Organization (WHO), it is the second leading cause of death worldwide, responsible for approximately 10 million annual fatalities. Future projections estimate a 63% rise by 2040, potentially resulting in 16.3 million deaths [1]. Cancer, a prevalent disease, arises from a combination of genetic mutations and epigenetic modifications [2]. Epigenetic modification controls, the process of regulating gene activation and inactivation in response to developmental and environmental signals, is mediated by epigenetic writers, erasers, and readers [3]. For instance, dysregulation of DNA methylation patterns has been observed across various cancer types, such as leukemia, breast, prostate, liver, and lung cancers [4]. Gene expression, governed by chromatin architecture and epigenetic regulation, is a fundamental determinant of cellular phenotype [5]. In eukaryotic cells, genomic DNA is packaged with histone proteins into chromatin, whose structural organization critically influences transcriptional activity [6]. Euchromatin, characterized by a relaxed configuration, permits gene transcription, whereas heterochromatin's dense packing typically represses expression [7].

Among various post-translational histone modifications, histone methylation plays a pivotal role in defining chromatin state and regulating gene accessibility [8]. This methylation is dynamically modulated by histone lysine methyltransferases (KMTs) and histone lysine demethylases (KDMs) [9]. KDM4 enzymes orchestrate epigenetic regulation of crucial oncogenes and tumor suppressor genes. KDM4C is an iron-dependent metalloenzyme belonging to the α-ketoglutarate-dependent dioxygenases with a Jumonji C (JmjC) domain, representing the second class of KDMs, and have emerged as a promising therapeutic target in oncology [10]. The KDM4 subfamily (KDM4A–F), notable members of the JmjC-domain demethylases, specifically demethylate tri- and di-methylated lysine residues on histone H3, particularly H3K9me3/me2 and H3K36me3, thereby influencing the transcriptional landscape of genes involved in cell proliferation and differentiation [11]. Among these, KDM4C (JMJD2C) has attracted attention due to its oncogenic potential. It selectively removes repressive methylation marks, such as H3K9me3, leading to chromatin decondensation and transcriptional activation of genes associated with tumor progression [12]. Furthermore, regulation of H3K36me3, typically associated with gene silencing, adds complexity to its role in epigenetic control. Overexpression or genomic amplification of KDM4C has been reported in multiple cancers, including prostate, breast, lung, and glioblastoma, correlating with enhanced cellular proliferation, invasion, and metastatic capacity [13]. For instance, in glioblastoma, KDM4C upregulation drives proliferation and tumorigenesis by activating c-Myc and suppressing p53 pro-apoptotic functions through demethylation of p53K372me1. Similarly, KDM4C contributes to senescence defense in TP53-mutated gastric cancer, and its silencing inhibits migration and enhances radio sensitivity in hepatocellular carcinoma [14]. Functional studies using CRISPR/Cas9 gene editing, siRNA-mediated silencing, and selective inhibitors such as SD70 have demonstrated that KDM4C suppression reduces tumor cell migration and metastasis in both in vitro and in vivo models [15]. The broad implication of KDM4 enzymes in oncogenesis, as seen in

prostate, breast, and colon cancers. The promising activity of KDM4 degraders like RDN8011 in esophageal cancer, underscores the therapeutic potential of targeting this histone demethylase family for various solid and hematological tumors. These findings support the therapeutic relevance of KDM4C and highlight its potential as a druggable epigenetic target [16]. Structural insights into the JmjC catalytic domain further facilitate rational drug design, establishing KDM4C as a promising candidate for the development of selective inhibitors aimed at disrupting oncogenic transcriptional programs in cancer therapy [17].

In parallel, natural compounds serve as invaluable scaffolds in cancer drug discovery due to their structural complexity and potent bioactivity, often exhibiting enhanced selectivity and reduced toxicity [18]. Various natural inhibitors target KDM4 enzymes reported [19]. Flavonoids [20] such as epigallocatechin, baicalein, and myricetin [21] chelate $Fe^{2+}$, potentially inhibiting other 2-OG dioxygenases [22]. Toxoflavin, a toxin from *Burkholderia gladioli*, effectively binds KDM4A, stabilizing it and reducing cell proliferation, with an $IC_{50}$ of 2.5 µM in cancer models [23]. In silico approaches have revolutionized this process by enabling rapid identification and optimization of lead candidates through computational techniques [24–26]. By integrating virtual screening [25], molecular docking, and dynamics simulations, researchers can efficiently predict drug-target interactions, assess pharmacokinetics, and prioritize compounds for synthesis [26]. This structure-based strategy not only reduces the cost and time associated with traditional drug development [27] but also facilitates the rational design of therapeutics targeting key oncogenic and epigenetic regulators [28]. This study systematically explores KDM4C as a druggable epigenetic target, using structural biology and computational approaches to identify novel inhibitors. By integrating high-resolution JmjC domain analysis with insilico screening, we aim to develop selective KDM4C inhibitors, advancing precision oncology and novel cancer therapies.

## 2. Materials and methods

The preparation of the protein and the ligand is crucial for ensuring stability during docking analysis.

### 2.1. Protein preparation

The crystal structure of KDM4C (PDB ID: 5KR7) was retrieved from the Protein Data Bank [29] prepared using Maestro's Protein Preparation Wizard under standard settings [30]. All heteroatoms, including crystallographic water molecules and ligands, were removed except catalytic site iron atom. Missing hydrogen atoms were added, and bond orders were assigned appropriately. Protonation states of ionizable residues were optimized at physiological pH using PROPKA, followed by hydrogen-bond network optimization. The structure was then subjected to restrained energy minimization using the OPLS4 force field [31], with particular attention given to the optimization of side-chain conformations, ensuring a high-quality model suitable for downstream structure-based drug discovery [32].

### 2.2. Ligand preparation

Accurate chemical preparation is essential for reliable molecular docking and binding affinity predictions. Ligand structures, comprising approximately 3673 natural organic compounds and flavonoids **(selleckchem.com)** were processed using the LigPrep module in Maestro (Schrödinger) [33]. Epik was employed to generate ionization states and relevant tautomers at physiological pH ($7.0 \pm 2.0$) [34]. Additionally, possible stereoisomers and ring conformations were enumerated to account for structural diversity. The generated conformers were subsequently energy-minimized using the OPLS4 force field, ensuring geometry optimization and elimination of high-energy structures. From the resulting set, the most stable conformer and stereoisomer of each compound were selected for subsequent molecular docking studies.

### 2.3. Receptor grid generation

Precise definition of the binding site is essential for reliable docking simulations. To this end, the Receptor Grid Generation tool in Glide (Schrödinger Suite) was utilized to construct a docking grid. The centroid of the co-crystallized ligand in

the KDM4C crystal structure (PDB ID: 5KR7) [29] was selected as the center of the grid box, ensuring alignment with the biologically relevant binding pocket. A grid box size of 15 Å was chosen to fully enclose the active site and accommodate ligand flexibility during docking [35]. The grid was optimized to balance computational efficiency and structural coverage of key residues involved in ligand recognition. Default parameters were applied for van der Waals scaling (scaling factor of 1.0) and partial charge cutoffs, ensuring a consistent framework for evaluating ligand-receptor interactions. This grid setup was instrumental for subsequent virtual screening, enabling accurate prediction of binding poses and interaction profiles.

## 2.4. Molecular docking

Molecular docking is a structure-based computational approach used to predict the preferred binding orientation and affinity of small molecules within the active site of a target protein, aiding in the identification of potential therapeutic candidates. In this study, docking was performed using the Glide module of Schrödinger's Maestro suite [36], applying a hierarchical virtual screening protocol. The screening pipeline included High-Throughput Virtual Screening (HTVS) for rapid filtering, Standard Precision (SP) for refined scoring, and Extra Precision (XP) for detailed pose prediction and affinity estimation [37]. HTVS efficiently processed a large chemical space, while SP and XP stages enhanced accuracy through improved sampling and scoring functions. Approximately 3,000 structurally diverse natural products and flavonoids, known for their pharmacological relevance, were screened against the KDM4C active site. Top-ranked compounds were selected based on binding energy and interaction profiles. This multi-step strategy provided a rigorous framework for identifying high-affinity ligands suitable for further structural and functional validation.

## 2.5. Molecular dynamics (MD) simulations

To investigate the dynamic behavior and stability of protein-ligand interactions at the KDM4C active site, molecular dynamics (MD) simulations were conducted using the Desmond [38]. This approach allowed for the evaluation of conformational flexibility and interaction stability over a 200 ns trajectory. The docked complex structures were solvated in a TIP3P water box [39] with a minimum edge distance of 10 Å. Further, we neutralized the systems by adding 0.15 M NaCl and minimized them by steepest descent and LBFGs algorithm with a maximum of 2000 iterations with convergence criteria of 1 kcal/mol/Å. The molecular dynamics production run, lasting 200 ns, was conducted for all energy-minimized complexes across eight distinct phases with specific constraints. The first seven phases encompassed equilibration and minimization to eliminate steric clashes and structure distortions, while the final phase involved extensive simulation. This prolonged production stage was maintained at 300 K using the Nosé–Hoover chain coupling scheme. The RESPA integrator facilitated the entire simulation process, ensuring precise bonding interaction calculations with a 2 fs time step. Electrostatic interactions were determined via the Particle Mesh Ewald (PME) algorithm, following a consistent protocol for all molecular dynamic's studies. The same protocol was employed for MD studies for all the complexes. Post-production run, trajectories were analyzed using RMSD, RMSF, and hydrogen bond evaluations, offering critical insights into ligand binding and complex dynamics, which are vital for drug discovery and development.

## 2.6. Binding free energy calculation

Molecular Mechanics Generalized Born Surface Area (MM-GBSA) calculations using Prime [40] were conducted to estimate the binding free energies of ligand-protein complexes, offering a more refined evaluation of binding affinities derived from molecular dynamics (MD) simulations [41]. This method combines molecular mechanics energies with solvation effects both polar and nonpolar and incorporates an entropy term [42] to provide a comprehensive measure of binding interactions [43]. The final 20 ns of equilibrated trajectories obtained from Desmond simulations were used for analysis. A total of 200 frames were extracted and processed using the thermal_mmgbsa.py script to compute energy components, including Coulombic and van der Waals interactions, solvation energies, and entropy contributions. These calculations

were performed for both ligand-bound complexes and control systems. Additionally, residue-wise decomposition enabled the identification of amino acids with significant energetic contributions to ligand stabilization [44].

$$\Delta G = \Delta E_{MM} + \Delta G_{solv} - T \cdot \Delta S$$

$$= \Delta E_{bat} + \Delta E_{vdW} + \Delta E_{coul} + \Delta, G_{solv,p}$$

$$+ \Delta G_{solv,np} - T \cdot \Delta S$$

The energy function in molecular mechanics (MM) simulations is composed of three primary terms: Ebat, EvdW, and Ecoul. The term Ebat encapsulates the contributions from bond stretching, angle bending, and torsional rotations within the molecular force field. EvdW represents the van der Waals interactions, accounting for non-bonded repulsive and attractive forces between atoms. Ecoul describes the electrostatic interactions, modelled through a Coulombic potential based on atomic charges. The solvation free energy is divided into polar and nonpolar components. The polar solvation free energy, denoted as Gsolv,p, is typically computed using the Generalized-Born (GB) model, which approximates the electrostatic effects of solvent molecules. The nonpolar solvation free energy, Gsolv, np, is generally estimated as a linear function of the solvent-accessible surface area (SASA), reflecting the hydrophobic interactions between the solute and solvent.

## 3. Results and discussion

KDM4C (Lysine Demethylase 4C) is a Fe (II) and α-ketoglutarate-dependent dioxygenase that plays a pivotal role in epigenetic regulation by catalyzing the demethylation of histone H3 at lysine residues 9 and 36 (H3K9me3 and H3K36me3) [45]. Notably, KDM4C exhibits substrate specificity, as it selectively demethylates the trimethylated forms of these residues while showing minimal to no activity toward mono- or dimethylated substrates, and it does not act on other histone marks such as H3K4, H3K27, or H4K20 [46]. Structurally, KDM4C possesses a highly compact active site, coordinated by a conserved Fe (II) ion that is critical for catalytic activity and overall protein stability [19]. The tightly constrained architecture of the active site presents a challenge for ligand accessibility, necessitating the identification of inhibitors capable of precise and stable interactions with the catalytic pocket [47] (**Fig 1A**). Dysregulated overexpression of KDM4C has been implicated in various malignancies, highlighting the therapeutic relevance of developing selective and potent inhibitors targeting this demethylase [48].

### 3.1. Molecular docking analysis

In the preliminary phase of molecular docking using the Glide module of the Schrödinger Suite, the protocol was validated by redocking the co-crystallized ligand from the KDM4C structure (PDB ID: 5KR7). This step ensured the reliability and precision of the docking methodology by assessing its ability to reproduce the experimentally observed binding pose. Glide's hierarchical virtual screening workflow, comprising High-Throughput Virtual Screening (HTVS), Standard Precision (SP), and Extra Precision (XP) modes, was then applied to systematically evaluate a curated compound library. HTVS mode enabled rapid screening of a large chemical space using shape-based algorithms, primarily to eliminate low-affinity ligands while maintaining computational efficiency. Compounds retained from this stage were subjected to SP docking, which integrates more refined scoring functions and increased sampling of ligand conformations. The final XP phase employed rigorous scoring, enhanced sampling, and comprehensive evaluation of molecular interactions, such as hydrogen bonding, hydrophobic contacts, and desolvation energies. This structured method enhanced predictive precision while reducing both false positives and negatives, ultimately benefiting ligands with strong binding specificity within the KDM4C

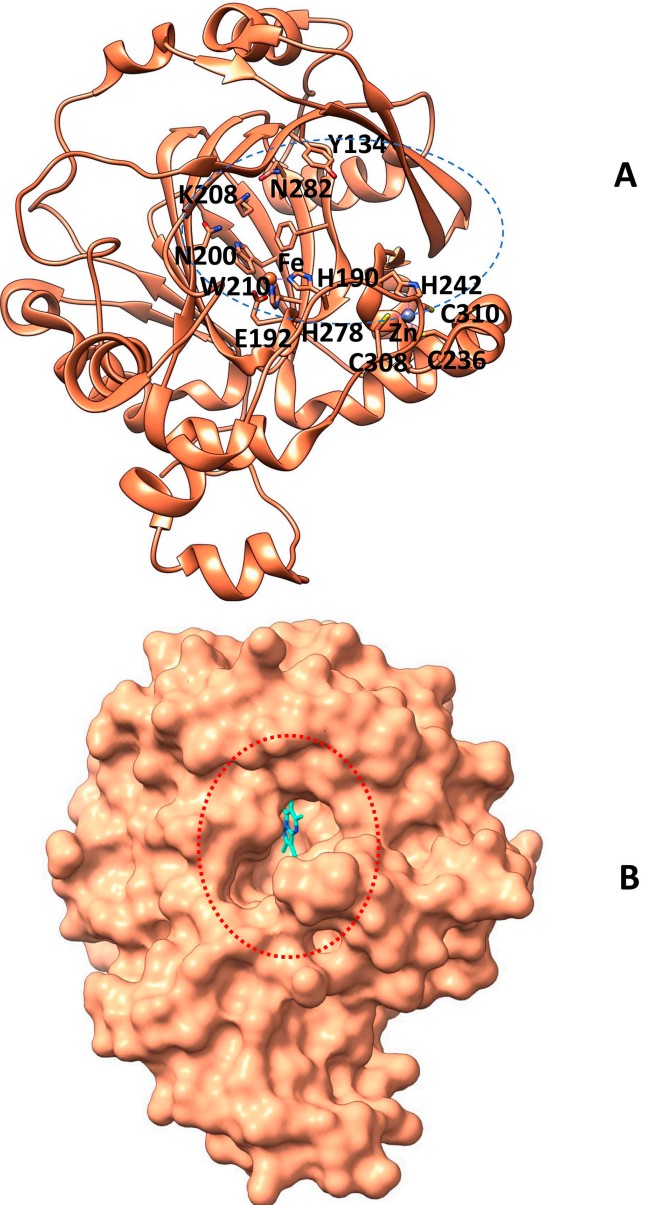

**Fig 1. Presented the three-dimensional (3D) structure of KDM4C (A) in a cartoon format, highlighting the primary catalytic residues as well.** **(B)** The comprehensive surface diagram of KDM4C illustrated a binding pocket encircled. The molecular graphics presented in the figures were created by UCSF Chimera **X.** The catalytic pocket is emphasized with a circular outline, while the ligand is depicted using a stick representation.

active site. The beneficial interaction features, thus facilitating the identification of potent demethylase inhibitors for later validation. These compounds, along with the control molecule, were subjected to comparative binding interaction studies.

### 3.2. Docking-based interaction analysis of the control molecule and top hits

Based on virtual screening, the top ten lead compounds and a control were shortlisted for detailed structural and comparative binding interaction analysis. The control compound, a bicyclic scaffold featuring one hydroxyl group, two secondary

amine (–NH) moieties, and one primary amine (–NH$_2$) substituent, demonstrated favourable binding within the KDM4C active site. Glide docking analysis revealed the formation of two key hydrogen bonds, one with Lys243 and another with Thr183, contributing to the anchoring of the ligand within the catalytic cleft. In addition to polar contacts, several stabilizing hydrophobic interactions were observed with surrounding residues, including Tyr134, Gly135, Ala136, Asp137, Thr179, Phe180, Gly181, Thr185, Phe187, and Met244 (**Fig 2A**). These interactions suggest a well-defined binding orientation and high complementarity between the control ligand and the binding site. The robustness of this interaction network supports the reliability of the docking protocol and validates its application for the subsequent screening and evaluation of structurally diverse ligands. Interaction profiles and conserved residue contacts were systematically considered in the comparative docking analysis.

Notably, the majority of the lead compounds exhibited binding patterns and key molecular interactions similar to those observed with the control molecule across the respective protein–ligand complexes. Among these, compound 707 (**Fig 2D**) was identified as the smallest in molecular size, potentially contributing to its distinct binding profile. Especially, compound 202 (**Fig 2C**) exhibited strong binding affinity, forming multiple stabilizing interactions within the active site of the KDM4C protein. Detailed docking analysis revealed the presence of four key hydrogen bonds involving the hydroxyl groups of the ligand and the side chains of Asn88, Lys243, Ser290, and Asn292. In addition, extensive hydrophobic interactions were observed between the ligand's aliphatic and aromatic carbons and the hydrophobic residues Tyr134, Tyr177, and Phe179, contributing to the overall stabilization of the complex (**Fig 3A**, **3C**). These specific interactions indicate a favorable binding orientation and suggest that compound 202 may serve as a promising candidate for further molecular dynamics and free energy evaluations. Ligand 707 (Fig 2D), characterized by a fused bicyclic scaffold bearing multiple hydroxyl substituents, demonstrated a well-defined binding conformation within the KDM4C active site (Fig 3D, 3F). It established two key hydrogen bonds with residues Arg311 and Tyr134, while simultaneously engaging in stabilizing van der Waals interactions with Tyr134, Lys242, and Lys283 (Fig 3D, 3F). Orientin, a tetracyclic polyphenolic flavonoid, formed three directional hydrogen bonds with Tyr87, Tyr134, and His278, and exhibited additional hydrophobic contacts with Tyr134, Phe187, and Tyr179 (Fig 4A, 4C). Oroxin, a pentacyclic polyphenolic compound, participated in five hydrogen bonds involving Tyr134, Ser290, Lys243, and Asn292, complemented by extensive van der Waals interactions with Tyr179 and Phe187 (Fig 4D, 4F). Similarly, Pectolinarin, another pentacyclic flavonoid, engaged Glu192, Asp193, and Asn292 via hydrogen bonding, while interacting with nonpolar residues such as Ala136, Tyr134, Val173, Tyr179, and Phe187 through hydrophobic contacts (Fig 5D, 5F). Among all compounds, Ligand 707 exhibited the most favorable binding energy at −10.11 kcal/mol, followed by compound 202 at −9.59 kcal/mol, Pectolinarin at −9.46 kcal/mol (**Table 1**). Notably, the 707-KDM4C complex displayed a highly stable binding mode, in contrast to compound 202, where steric clashes and unfavorable contacts likely contributed to conformational instability and reduced binding efficiency.

### 3.3. ADMET evaluation

ADMET analysis is essential in evaluating the pharmacokinetic profile and safety of natural compounds in drug development. The selected top natural compound adheres to Lipinski's Rule of Five, exhibiting acceptable molecular weight, lipophilicity (logP), and numbers of hydrogen bond donors and acceptors, suggesting good oral bioavailability. Solubility, Caco-2 permeability, and blood-brain barrier penetration parameters also fall within acceptable ranges, indicating favourable absorption and distribution. Toxicity prediction shows a high LD50 value and low toxicity class, implying minimal risk. Overall, the compounds (202, 707, orientin, Oroxin, and Pectolinarin) demonstrates promising drug-likeness, supporting its potential as a safe and efficacious candidate for further preclinical evaluation (Tables 2 and 3).

### 3.4. MD simulation analysis

By evaluating parameters such as docking score, Glide Emodel, Glide energy, binding mode relative to the positive control, and structural diversity, we shortlisted the top five complexes, i.e., 202, 707, Orientin, Oroxin, and Pectolinarin.

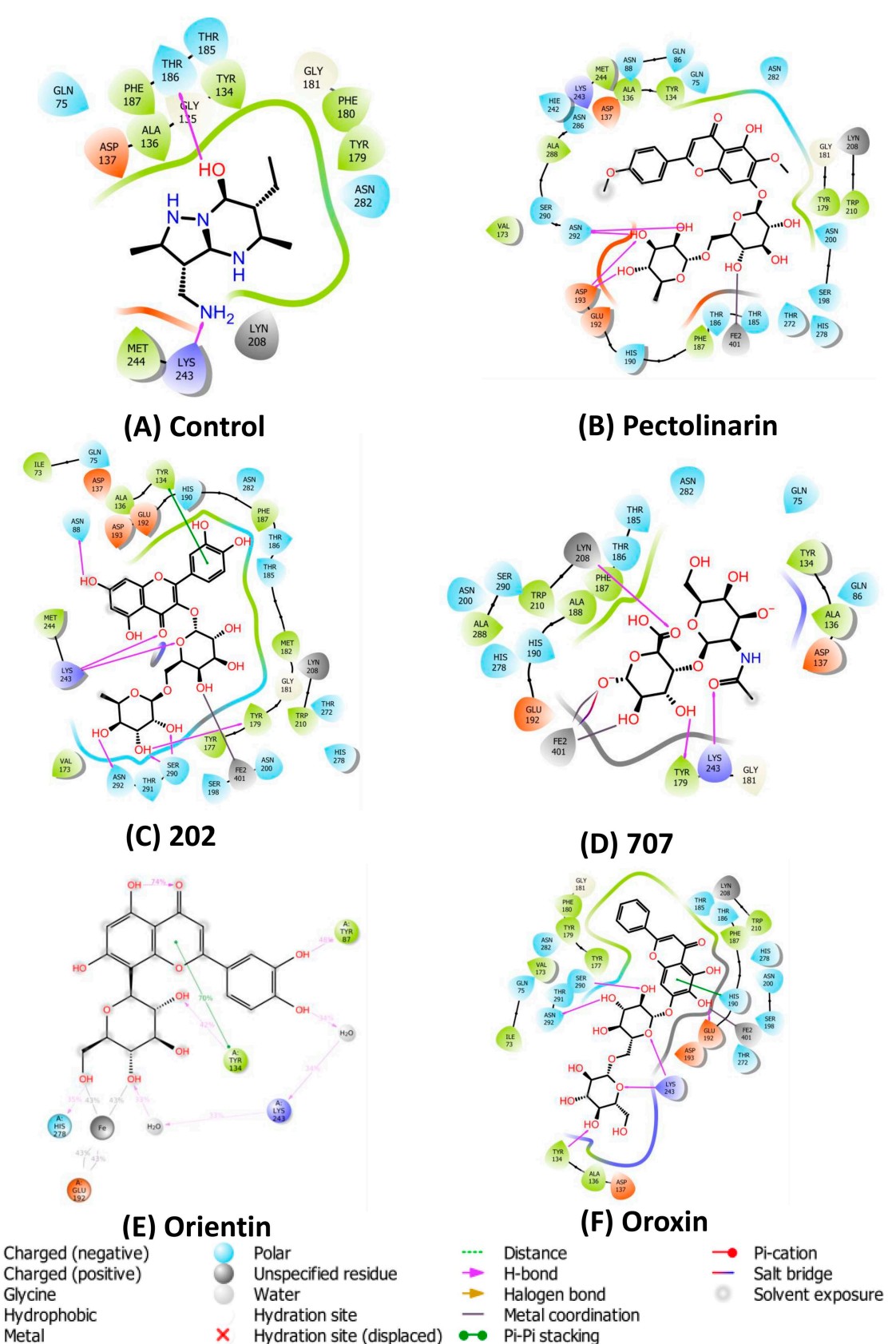

**(A) Control**

**(B) Pectolinarin**

**(C) 202**

**(D) 707**

**(E) Orientin**

**(F) Oroxin**

Charged (negative) — Polar — ---- Distance — •— Pi-cation
Charged (positive) — Unspecified residue — ➤ H-bond — —— Salt bridge
Glycine — Water — ➤ Halogen bond — Solvent exposure
Hydrophobic — Hydration site — —— Metal coordination
Metal — ✕ Hydration site (displaced) — ●—● Pi-Pi stacking

**Fig 2. The illustration depicts the two-dimensional interactions between ligands and proteins, highlight the residues located at the ligand-binding site. (A)** illustrate the interactions between the control ligand and KDM4C, **(B)** depicts the interactions of ligand1 (Pectolinarin) with KDM4C, **(C)** outlines the interactions of ligand2 (202) with KDM4C, **(D)** details the interactions of ligand3 (707) with KDM4C, **(E)** presents the interaction of ligand4 (Orientin) with KDM4C, and **(F)** highlights the interactions of Oroxin with KDM4C.

Molecular dynamics simulations were then carried out to assess the stability and interactions of these compounds in dynamic system: control, 202, 707, orientin, Oroxin, and Pectolinarin. Among these, complexes Pectolinarin and 202 demonstrated greater stability followed by Orientin, compared to the positive control during simulations. As a result, our study emphasized these two complexes along with the control molecule for further analysis and exploration.

### 3.5. RMSD and RMSF analysis

To calculate RMSD, protein frames are aligned to the reference frame backbone, and RMSD is computed based on atom selection. Stabilized fluctuations indicate equilibration near a thermal average structure. For small proteins, changes of 1–3 Å are acceptable, while larger shifts imply significant conformational alterations. RMSF measures local fluctuations, with peaks highlighting dynamic regions like terminal tails, while rigid structures like alpha helices and beta strands show minimal changes compared to loops. The Root-Mean-Square Deviation (RMSD) was analysed separately for the protein and the ligand to evaluate the stability of the complex and the conformational dynamics of the bound ligand throughout the molecular dynamics simulation. The protein backbone RMSD across all simulated complexes demonstrated stable trajectories, with fluctuations consistently below 3.0 Å. Most top-ranked complexes protein maintained an optimal RMSD range of 0.5–1.0 Å following the pre-equilibration phase (Fig 6A). Ligand RMSD analysis, however, revealed distinct stability profiles (Fig 6B). Pectolinarin exhibited exceptional stability, maintaining an RMSD below 1 Å throughout the entire 200 ns simulation. Similarly, ligand 202 stabilized after an initial deviation at 30 ns, fluctuating minimally around 4 Å for the remainder of the simulation. In contrast, orientin and compound 707 displayed significant deviations, particularly during the 40–60 ns frame, before attaining more consistent trajectories. The remaining compounds, including the protein-control complex, showed RMSD values exceeding 5 Å, indicating substantial conformational flexibility and reduced binding stability. The protein-control complex itself achieved an average RMSD of 6 Å, stabilizing between 4–5 Å after 60 ns. Based on the RMSD analysis, Pectolinarin and ligand 202 were identified as forming the most stable complexes, characterized by low RMSD values and minimal fluctuations. In contrast, the remaining ligands exhibited significant conformational instability, indicative of poor binding. Consequently, only Pectolinarin and ligand 202 were advanced to the final round of analysis. Further analysis of the root mean square fluctuation (RMSF) revealed that all compounds exhibited stable trajectories, with lower fluctuation values observed across the protein residues in comparison to other compounds (Fig 7). As expected, the N- and C-terminal regions of the protein demonstrated higher fluctuations, consistent with their inherent flexibility, whereas the secondary structural elements, such as α-helices and β-strands, exhibited greater rigidity and relatively lower fluctuations. These regions were notably more stable than the loop regions, which are typically less structured. Similar patterns of fluctuation were observed across all complexes, particularly in the residue ranges 140–150, 160–175, 220–230, and 290–310. These fluctuations are attributed to the presence of undefined or irregular secondary structural elements within these regions.

### 3.6. Binding interaction analysis

**Time dependent hydrogen bond interactions.** The stability and integrity of protein-ligand complexes are significantly influenced by their ability to form and sustain intermolecular hydrogen bonds (H-bonds). To evaluate this aspect, we analyzed the time evaluation of intermolecular H-bonds throughout the simulation to assess the stability of the KDM4C-ligand complexes. The results revealed that the Pectolinarin complex consistently maintained 5–7 hydrogen bonds during

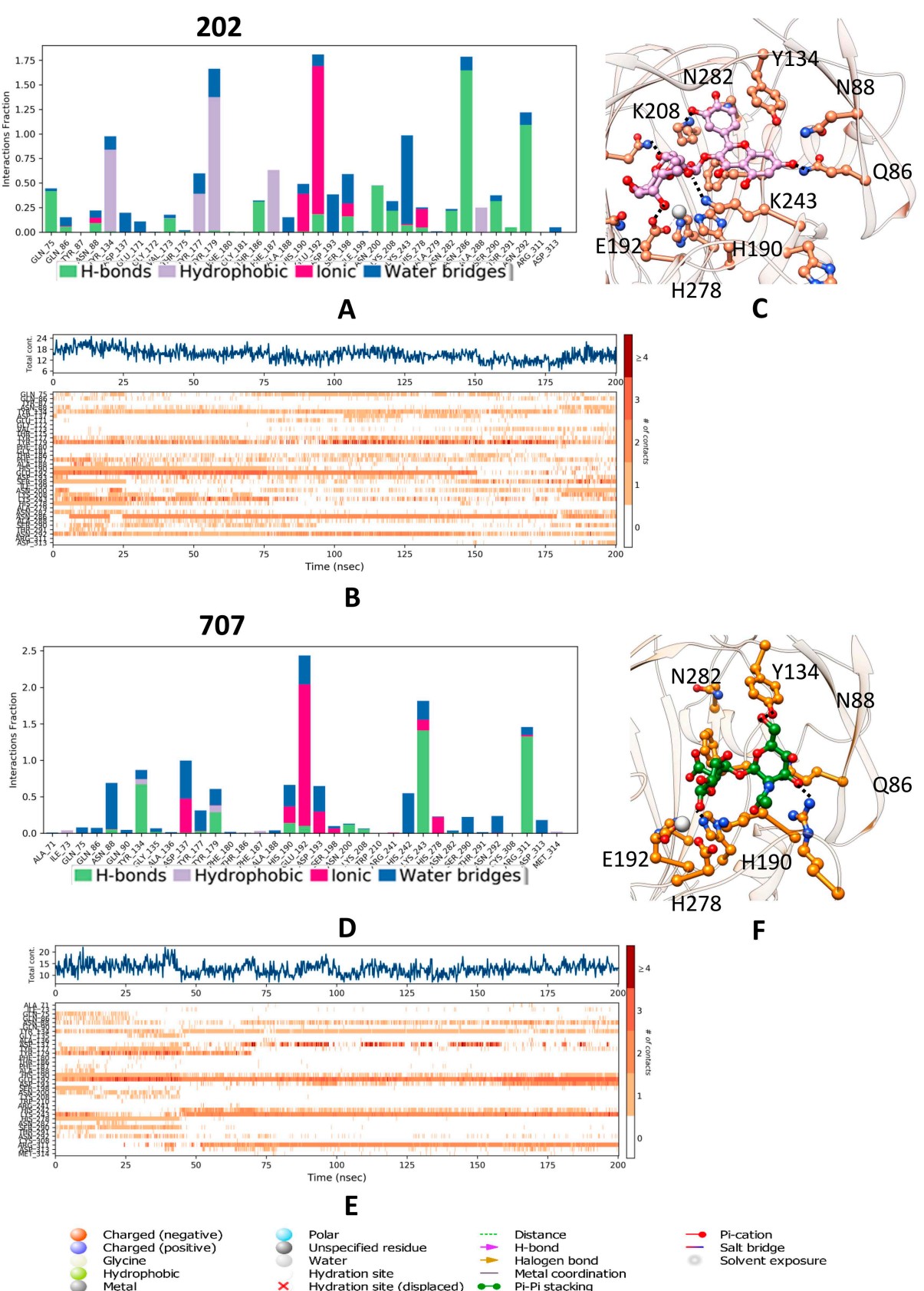

**Fig 3. Molecular dynamics analysis of KDM4C interactions with the ligand 202 and 707 (A) A histogram illustrate the interaction fractions with active site amino acid residues and (B) A timeline depiction of the simulated complex, highlighting interactions with all amino acid residues at each time point, where multiple specific contacts with the ligand (202) are indicated by a darker shade. (C)** represents the interaction of 202 ligand with the active site residues of KDM4C. **(D)** Another histogram display interaction fractions with active amino acid residues. **(E)** A timeline representation of the simulated complex, again showing interactions with all amino acid residues at each time frame. **(F)** represents the interaction of 707 ligand with the active site residues of KDM4C.

the simulation period, reflecting strong interactions within the docked complex, as observed from the simulated trajectories (Fig 8A). Furthermore, the study indicated that Complex 202 exhibited stable hydrogen bonding interactions, with five to six H-bonds consistently observed throughout the simulation. In comparison, the Orientin complex demonstrated slightly fewer hydrogen bonds, averaging 3–4 bonds, while the Oroxin complex exhibited 4–5 hydrogen bonds during the simulation.

### 3.7. Interplay of residues and stability in dynamics

The simulation revealed that the positive control molecule formed 3–4 hydrogen bonds with residues Thr186, Asn88, Lys208, and Asn282, contributing to the stability of the complex. Water-mediated hydrogen bonds were observed with Asp137, Ala188, and Asn286, further reinforcing the structural integrity. Stabilization was additionally supported by van der Waals interactions involving key residues Tyr134, Tyr179, Phe187, and Trp210. These interactions collectively indicate a robust binding mechanism and dynamic stability for the positive control molecule.

Pectolinarin establishes six to seven hydrogen bonds, including water-mediated interactions, with key residues Asp193, Lys208, Asn282, Asp137, Glu192, and Asn292, contributing to the stabilization of the complex during dynamic motion. Additionally, two major salt bridges and one partial salt bridge further enhance the stability of the complex, involving residues Glu192, His190, and partially His278. Several Van der Waals interactions are also identified, engaging residues Tyr134, Ala136, Asp137, Tyr179, Gly181, Phe187, Ser198, Asn200, Trp210, Asn286, and Ala288, collectively stabilizing the interaction. This interaction analysis is visually presented in the accompanying diagram, where the timeline representation effectively illustrates binding events, while the histogram further validates the interactions observed in the dynamic system.

In complex 202, interaction analysis reveals the formation of 5–6 hydrogen bonds, including water-mediated interactions, with key residues Gln75, Glu192, Asn200, Asn282, Asn286, Ser290, and Asn292, which collectively contribute to the stabilization of the complex during dynamic motion. Water-mediated interactions were also observed with residues Asp193, Ser198, and Lys243, further reinforcing the stability of the system. Salt bridges involving His190, Glu192, and partially His278 play a significant role in the stabilization of the complex. Additionally, van der Waals interactions with residues Tyr174, Tyr177, Tyr179, Phe187, and Ala288 provide supplementary support to the integrity of the complex. This comprehensive interaction analysis is visually summarized in the accompanying diagram, with the timeline representation effectively illustrating binding events. A histogram further corroborates these observations, emphasizing the dynamic stability of the interactions.

The simulation demonstrated that the Oroxin complex formed 5–6 hydrogen bonds with residues Tyr134, Lys243, Glu192, Ser290 and Asn292, including water-mediated interactions with Asp137, His190, Lys243, and Asp313. Stabilization was further supported by van der Waals interactions involving Tyr179 and Phe187. However, the absence of salt bridges indicated suboptimal structural stability. In comparison, Complex 707 exhibited only 2–3 hydrogen bonds with residues Tyr134, Lys243, and Arg311, along with water-mediated bonds involving Asn88, His242, Asp193, and Glu192. Salt bridges involving Asp137 and Glu192 were unstable and sparsely distributed, further underscoring limited stability. Both complexes demonstrated inferior stability relative to the earlier complexes, as evidenced by interaction analysis.

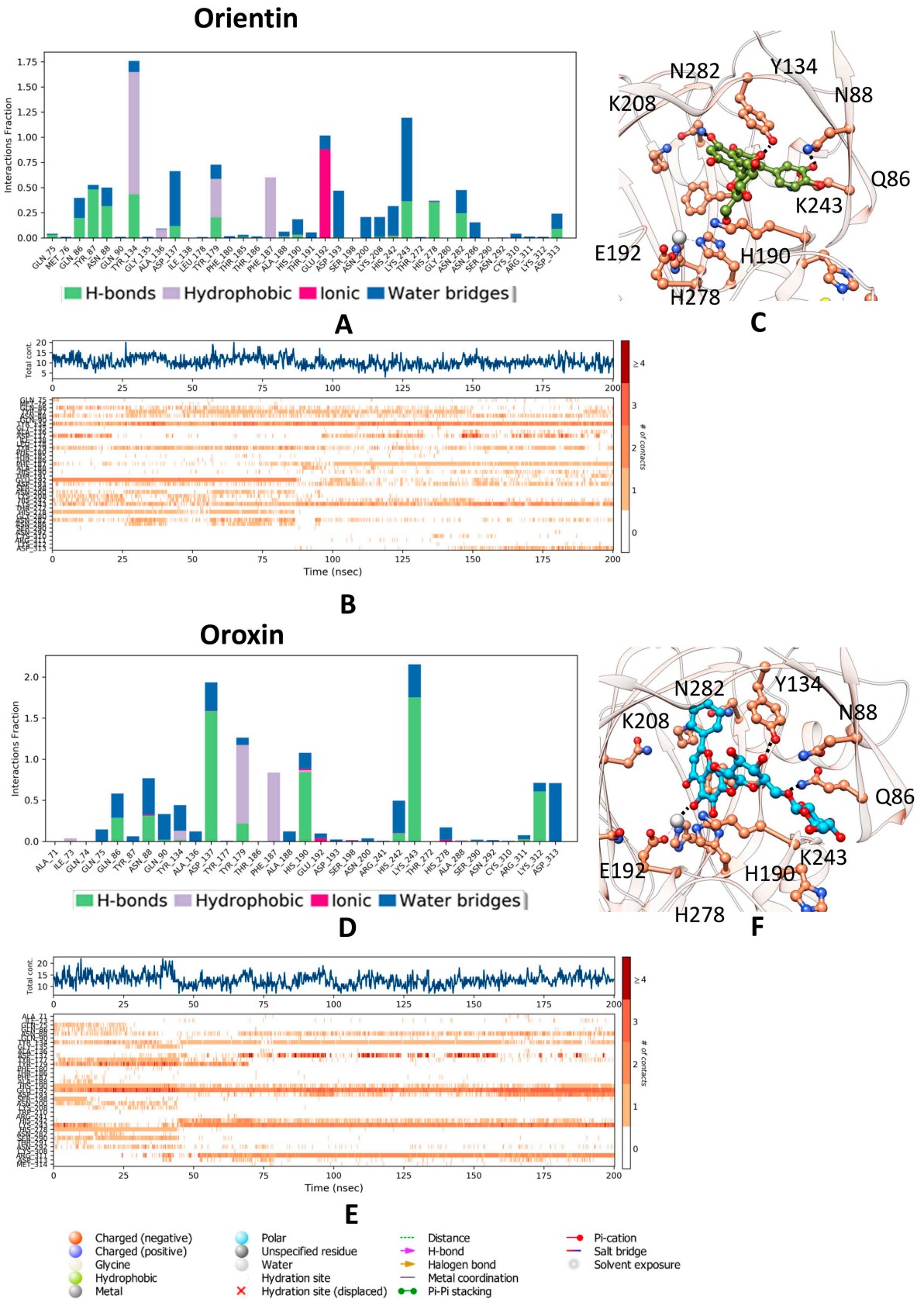

**Fig 4. Molecular dynamics analysis of KDM4C interactions with the Orientin and Oroxin: (A) A histogram illustrating the interaction fractions with active site amino acid residues. (B)** A timeline depiction of the simulated complex, highlighting interactions with all amino acid residues at each time point, where multiple specific contacts with the ligand (202) are indicated by a darker shade. **(C)** represents the interaction of Orientin ligand with the active site residues of KDM4C. **(D)** Another histogram displaying interaction fractions with active amino acid residues. **(E)** A timeline representation of the simulated complex, showed interactions with all amino acid residues at each time frame. **(F)** represents the interaction of Oroxin ligand with the active site residues of KDM4C.

## 3.8. Radius of gyration (Rg)

The Rg derived from MD simulations is a critical parameter for evaluating the compactness and stability of a protein-ligand complex. Rg measures the distribution of atomic positions relative to the protein's center of mass, providing insights into its conformational dynamics. A decrease in Rg indicates ligand-induced compaction, potentially stabilizing a favorable conformation for therapeutic action, whereas an increase may signify destabilization. Monitoring Rg over simulation time offers valuable information on ligand-induced structural changes and complex stability.

In the graph (Fig 8B) illustrates Rg fluctuations over 200 ns. Among the tested compounds, Pectolinarin (red) emerges as the most effective, demonstrating transient compaction with dips near 20.0 Å. Orientin (green) exhibits stable Rg values (20.3–20.5 Å), akin to the control (blue), highlighting consistent compactness. Complex 202 (grey) ranks third with moderate fluctuations, while Oroxin (pink) and Complex 707 (light green) show limited stability with inconsistent Rg values.

## 3.9. Post dynamics binding free energy calculation

The interaction strengths between the selected ligands and the KDM4C protein were quantitatively assessed using the MM/GBSA free energy method. This post-processing approach leverages Molecular Mechanics (MM) force fields combined with the Generalized Born Surface Area (GBSA) continuum solvation model to estimate the binding free energy ($\Delta$Gbind). Specifically, $\Delta$Gbind was calculated by evaluating the free energy of the protein-ligand complex, the unbound protein, and the unbound ligand in solution, incorporating contributions from molecular mechanics energies (bond, angle, dihedral, van der Waals, and electrostatic), polar and nonpolar solvation terms, and an estimated entropic component. MM/GBSA calculations were performed on an ensemble of representative conformations extracted from the final 20 ns of Molecular Dynamics (MD) trajectories for five compounds: 207, 707, Orientin, Oroxin, and Pectolinarin, alongside a co-crystallized ligand control. The calculated $\Delta G_{bind}$ values were further decomposed into individual energy terms (e.g., $\Delta E_{elec}$, $\Delta E_{vdw}$, $\Delta_{Gpolar}$, $\Delta G_{nonpolar}$) to elucidate the energetic contributions driving ligand recognition. The results reveal a binding affinity hierarchy: Pectolinarin ($\Delta G_{bind}$ = −57.33 kcal/mol), Compound 202 ($\Delta G_{bind}$ = −57.21 kcal/mol)> Orientin ($\Delta G_{bind}$ = −55.36 kcal/mol)> Oroxin ($\Delta G_{bind}$ = −47.58 kcal/mol)> Compound 707 > Control (**detailed in the Table 4**). The negative $\Delta G_{bind}$ values across all ligands indicate thermodynamically favorable binding to KDM4C. The observed trend in binding affinities provides a quantitative framework for prioritizing Pectolinarin and compound 202 as high-affinity ligands for subsequent structural, biochemical, and pharmacological evaluation in the context of KDM4C inhibition.

## 3.10. Per-residue energy decomposition

Per-residue energy decomposition analysis using the MM-GBSA approach performed to evaluate the contribution of individual amino acids to ligand binding in the KDM4C-ligand complexes. This analysis identified key residues that play a crucial role in stabilizing protein-ligand interactions. By breaking down binding energy at the residue level, the study highlighted critical interactions, including hydrogen bonds, van der Waals forces, and electrostatic contributions. Residues within the active site exhibited notably negative free energy values, underscoring their importance in ligand affinity shown in **Fig 9**. These findings provide valuable mechanistic insights into binding interactions and offer a foundation for further refinement and optimization of lead compounds [49].

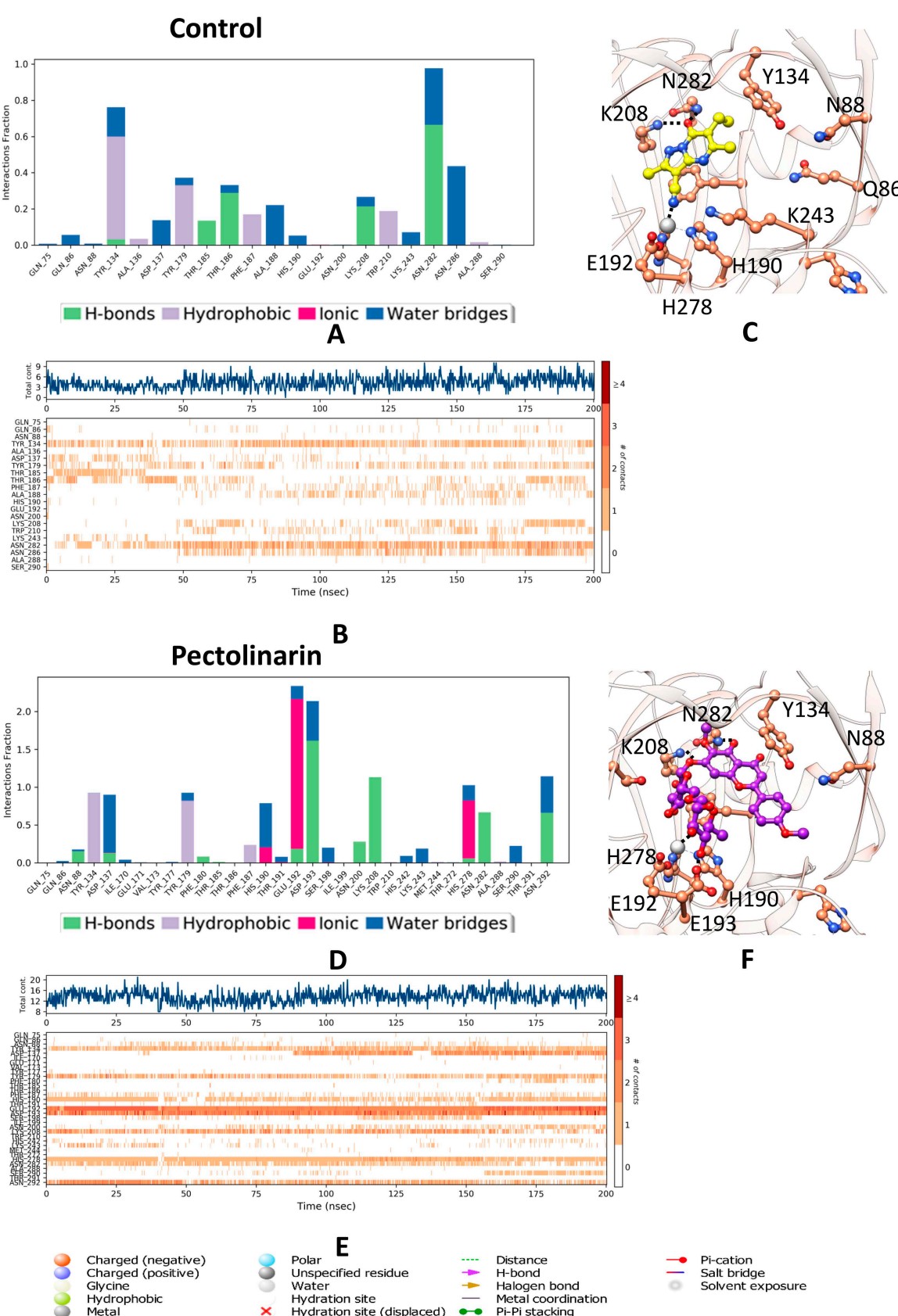

**Fig 5. Molecular dynamics analysis of KDM4C interactions with the control and Pectolinarin (A) A histogram illustrating the interaction fractions with active amino acid residues and control ligand. (B)** A timeline depiction of the simulated complex, highlight interactions with all amino acid residues at each time point, where multiple specific contacts with the control ligand are indicated by a darker shade. **(C)** represents the interaction of control ligand with the active site residues of KDM4C. **(D)** Another histogram display the interaction fractions with active amino acid residues. **(E)** A timeline representation of the simulated complex, again show interactions with all amino acid residues at each time frame. **(F)** represents the interaction of Pectolinarin with the KDM4C.

**Table 1. Post docking and binding energy analysis.**

| S.no | Ligand | Docking Sore (kcal/mol) | Ligand efficiency | Glide emodel | Glide Energy | Kd |
|---|---|---|---|---|---|---|
| 1 | Control | −7.03 | −0.27 | −105.86 | −53.285 | $3.39 \times 10^{-8}$ |
| 2 | Pectolinarin | −9.46 | −0.42 | −68.35 | −50.34 | $1.67 \times 10^{-8}$ |
| 3 | 202 | −9.59 | −0.44 | −77.83 | −55.84 | $3.07 \times 10^{-9}$ |
| 4 | 707 | −10.11 | −0.40 | −81.97 | −57.39 | $4.82 \times 10^{-9}$ |
| 5 | Orientin | −9.05 | −0.44 | −74.56 | −53.73 | $8.49 \times 10^{-9}$ |
| 6 | Oroxin | −9.34 | −0.43 | −79.35 | −56.62 | $1.00 \times 10^{-8}$ |

**Table 2. QIK PROP (ADMET).**

| Ligand | Molecular Weight (Da) | log BB | Rotatable Bonds | HBD/ HBA | log P (O/W) | HERK (log $IC_{50}$) | log Kp | SASA | Caco2 | MDCK | Oral Absorption (%) |
|---|---|---|---|---|---|---|---|---|---|---|---|
| Control | 228 | −5.51 | 2 | 4/5 | 2.45 | −4.20 | −7.61 | 902 | 21 | 11 | 30 |
| Pectolinarin | 357.4 | −1.97 | 7 | 2/7 | −0.91 | −1.59 | −0.234 | 639.3 | 757 | 917 | 93 |
| 202 | 610.5 | −1.65 | 6 | 16/10 | 1.58 | −2.30 | −10.26 | 635 | 66 | 53 | 61 |
| 707 | 397.3 | −1.11 | 6 | 12/8 | −1.26 | −5.81 | −11.91 | 636.1 | 481 | 224 | 91 |
| Orientin | 448.3 | −2.05 | 3 | 11/8 | 1.27 | −6.27 | −9.14 | 631.1 | 233 | 102 | 86 |
| Oroxin | 594.5 | −2.64 | 7 | 7/15 | 1.90 | −4.95 | −10.16 | 563.4 | 144 | 62 | 72 |

**Table 3. Toxicity prediction of compounds.**

| Ligands | Hepatotoxicity | Nephrotoxicity | Neurotoxicity | Cytotoxicity | Carcinogenicity | BBB Barrier | Immunotoxicity | LD50(mg kg$^{-1}$)/Class |
|---|---|---|---|---|---|---|---|---|
| **Control** | Inactive | Active | Active | Inactive | Active | Active | Inactive | 3000/5 |
| | 0.62 | 0.51 | 0.62 | 0.73 | 0.58 | 0.76 | 0.99 | |
| **Pectolinarin** | Inactive | Active | Inactive | active | Inactive | Inactive | Active | 5000/5 |
| | 0.85 | 0.76 | 0.87 | 0.55 | 0.93 | 0.96 | 0.99 | |
| **202** | Inactive | Active | Inactive | Inactive | Inactive | Inactive | Active | 5000/5 |
| | 0.87 | 0.77 | 0.89 | 0.64 | 0.91 | 0.75 | 0.98 | |
| **Orientin** | Inactive | Active | Inactive | Inactive | Inactive | Inactive | Active | 1213/4 |
| | 0.81 | 0.67 | 0.88 | 0.87 | 0.72 | 0.55 | 0.52 | |
| **Oroxin** | Inactive | Active | Inactive | Inactive | Inactive | inactive | Inactive | 4000/5 |
| | 0.83 | 0.78 | 0.74 | 0.67 | 0.87 | 0.61 | 0.85 | |
| **707** | Inactive | Active | Inactive | Inactive | Inactive | Inactive | Inactive | 800/4 |
| | 0.67 | 0.81 | 0.58 | 0.68 | 0.72 | 0.93 | 0.94 | |

## 4. Discussion

Targeting KDM4C is particularly relevant in cancer, as histone methylation dynamically regulates gene expression involved in cell proliferation and survival. Demethylation of H3K9 and H3K36 by enzymes like KDM4C can activate oncogenic transcriptional programs [50]. Overexpression of KDM4C and related demethylases has been linked to increased

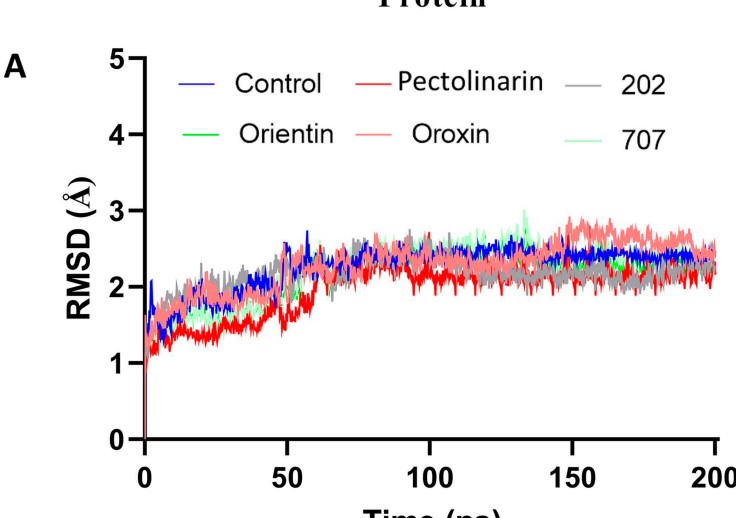

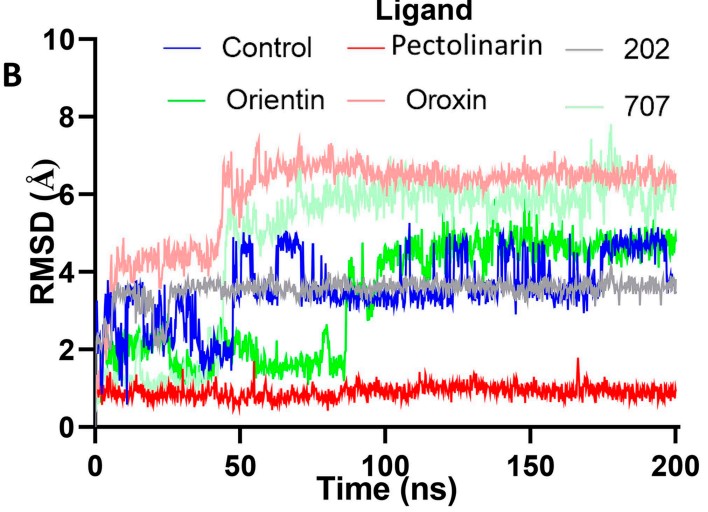

**Fig 6. Represents the stability of protein and protein-ligand interactions during the 200 ns MD simulation. (A)** showed the RMSD values for protein backbone and ligand fluctuations during the simulation. Figures **(B)** show the root-mean-square fluctuation (RMSF) in the general movement of each residue of protein-ligand complexes such as control, Pectolinarin, 202, 707, Orientin and Oroxin ligands during the total simulation time.

tumor growth, invasion, and metastasis in various cancers [51]. Thus, inhibiting KDM4C offers a promising strategy for restoring epigenetic control and suppressing cancer progression [52]. Hence this study employs a structure-based drug discovery approach to identify natural compounds as potential inhibitors against the JmjC domain of KDM4C, a histone demethylase, through an integrative computational pipeline molecular docking, molecular dynamics (MD) simulations, MM/GBSA free energy calculations, and ADMET profiling we identified natural compounds with high binding affinity and dynamic stability within KDM4C's catalytic pocket. These compounds engage critical residues (His190, Glu192, and His278) in the JmjC domain, disrupting Fe (II) and 2-oxoglutarate (2-OG) coordination essential for H3K9me3/H3K36me3 demethylation, thus inhibiting oncogenic transcription similar to other studies [53]. This approach has been widely adopted

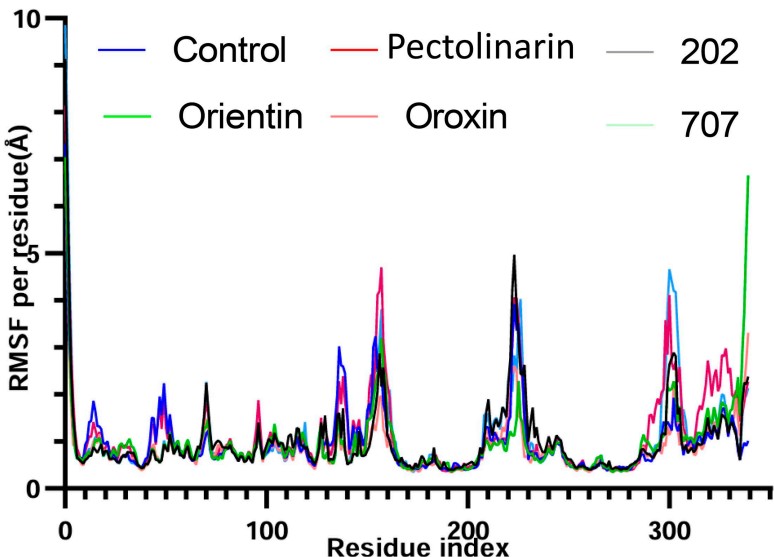

**Fig 7. Represents the stability of protein-ligand interactions during the 200 ns MD simulation. (A)**, RMSD values for protein backbone and ligand fluctuations during the simulation **(B)** represent the RMSD of the ligand's fluctuations with the protein.

in numerous studies to discover novel inhibitors, underscoring its efficacy in structure-based drug discovery. While molecular docking offers a useful initial screening of protein–ligand interactions, it has inherent limitations due to simplified scoring functions and static models [54]. To provide a more comprehensive assessment, we complemented docking along with 200 ns molecular dynamics (MD) simulations, which confirmed the stability of the top compounds within the KDM4C catalytic pocket in dynamic condition. It further supported by stable RMSD and compact Rg values. Additionally, MM/GBSA binding free energy analysis indicated favorable interactions, largely driven by van der Waals and electrostatic forces [55]. Per-residue energy decomposition further highlighted key active site residues His190, Glu192, and His278 as major contributors to ligand binding, aligning with their established roles in KDM4C catalysis. Key inhibitors, such as caffeic acid (IC$_{50}$: 13.7 µM for KDM4C), exhibit dual inhibition of KDM4C and KDM6A [56], with demonstrated antiproliferative effects in oesophageal squamous cell carcinoma [57]. The favorable dietary composition and strong safety profile enhance its suitability for practical use. Similarly, SD70 exhibited [56] an IC$_{50}$ range of 1–30 µM in various Cancer Cell lines, effectively inhibiting KDM4C and other KDMs. Evidence from validation studies showed a decrease in tumor formation in xenograft models, further emphasizing the significance of natural scaffolds as epigenetic regulators. Molecular docking revealed hydrogen bonding and pi-pi stacking with residues like Tyr177 and Lys241, stabilizing protein-ligand complexes. MD simulations (200 ns) confirmed conformational integrity (RMSD: 2.5–3.5 Å; stable Rg), while MM/GBSA calculations yielded favorable binding energies (e.g., −70.23 kcal/mol for caffeic acid), driven by van der Waals and electrostatic interactions. ADMET profiling affirmed the drug-likeness of all top five compounds that were screened, while also validating Lipinski's rule of five. The ADMET evaluation also indicated that these compounds exhibited high solubility and low toxicity computationally, which are essential for the pharmacological measurement of drug molecules. However, KDM4C's auxiliary domains (PHD, Tudor) suggest regulatory complexity, warranting future studies to enhance inhibitor specificity and minimize off-target effects [47]. These findings validate natural compounds as viable templates for KDM4C inhibition, guiding scaffold optimization for improved potency and selectivity. Experimental validation, including enzymatic assays and in vivo studies, is critical to advance these candidates toward clinical application, exemplifying the synergy of computational and experimental approaches in epigenetic drug discovery.

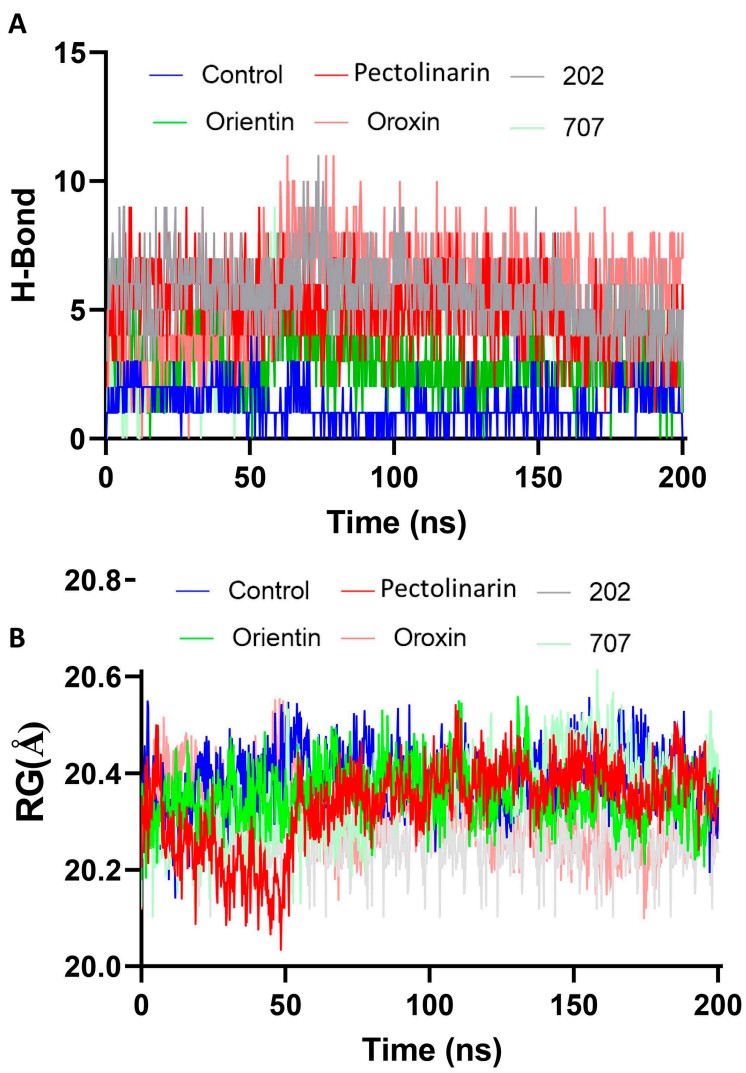

**A**

**B**

**Fig 8. Time evolution plot of hydrogen bonds and radius of gyration between the ligands and KDM4C. A)** showed the hydrogen bonds interaction between protein and ligand during the 200 ns time of simulation. **B)** Display the radius of gyration (Rg) for the protein-ligand interaction, as determined from the 200 ns simulation. The colors red, green, blue, grey, pink, and light green corresponds to the ligands Pectolinarin, Orientin, control, 202, Oroxin, and compound 707.

**Table 4. MMGBSA-energy fractions.**

| KDM4C Complexes | dG_Binding energy (kcal/mol) | dG_Bind_Coulomb energy (kcal/mol) | dG_Bind_Covalent energy (kcal/mol) | dG_Bind_Van der Waals energy (kcal/mol) | Lipophilic energy (kcal/mol) | ΔG_Bind_Solv_GB (kcal/mol) |
|---|---|---|---|---|---|---|
| Control | −28.83 ± 2.6 | −11.72 ± 1.5 | 0.88 ± 0.4 | −31.00 ± 2.4 | −15.34 ± 1.6 | 29.34 ± 1.7 |
| Pectolinarin | −57.33 ± 2.3 | −42.77 ± 1.7 | 11.60 ± 0.6 | −53.32 ± 0.5 | −23.69 ± 1.7 | 56.19 ± 0.7 |
| 707 | −36.07 ± 2.6 | −30.28 ± 0.5 | 0.44 ± 1.2 | −26.67 ± 3.1 | −3.02 ± 1.1 | 28.70 ± 1.4 |
| Orientin | −55.36 ± 2.2 | −30.74 ± 1.6 | 8.10 ± 0.8 | −48.77 ± 2.8 | −8.93 ± 0.9 | 31.18 ± 2.6 |
| Oroxin | −47.58 ± 1.8 | −55.75 ± 1.9 | 4.56 ± 0.6 | −58.82 ± 1.1 | −17.88 ± 2.1 | 64.29 ± 3.8 |
| 202 | −57.21 ± 2.8 | −46.90 ± 0.9 | 4.79 ± 0.9 | −71.94 ± 2.3 | −16.18 ± 1.7 | 87.73 ± 3.1 |

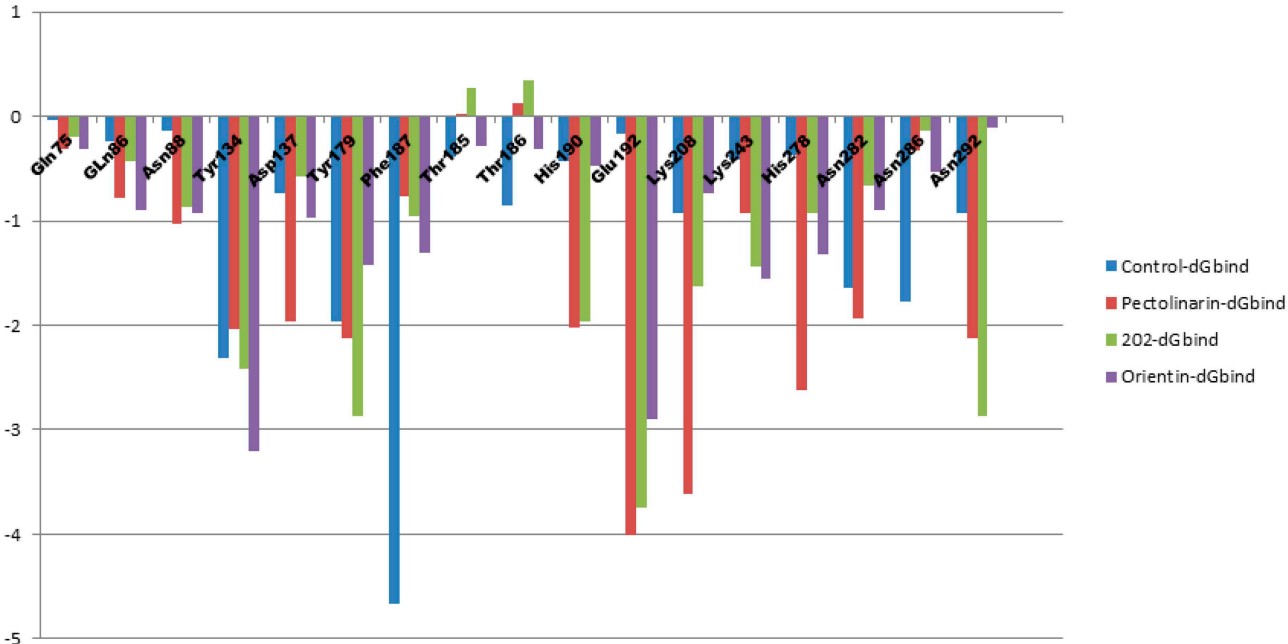

**Fig 9. A detailed analysis of the binding energy per residue for the top three complexes and control, showcasing the impact of specific amino acid residues on the total binding energy.** The graph emphasizes crucial residues that facilitate molecular interactions and enhance the stability of the complexes.

## 5. Conclusions

Our study highlights the therapeutic potential of natural polyphenolic compounds as inhibitors of KDM4C, an oncogenic epigenetic regulator implicated in cancers like breast, prostate, colorectal, and hepatocellular carcinoma. Through in silico high-throughput screening, we identified five candidates with superior binding affinity compared to the control ligand (6X9), demonstrating favorable structural complementarity, binding energetics, and stability within KDM4C's active site. These natural compounds, with their potential for fewer side effects compared to synthetic drugs, offer promising scaffolds for targeted cancer therapy. Since this computational study highlights promising natural polyphenolic compounds as potential KDM4C inhibitors for cancer therapy, several limitations must be addressed before clinical translation. Notably, our findings are based solely on in silico predictions, which do not account for the complexity of biological systems. Experimental validation is needed to confirm inhibitory potency, selectivity, safety, and pharmacokinetic properties like bioavailability and toxicity. Moreover, specificity is critical to avoid off-target effects due to KDM4C's role in multiple signaling pathways. Resistance mechanisms also need investigation. To advance these candidates, future work should prioritize in vitro biochemical and enzymatic assays to confirm binding and activity, followed by cell-based studies and animal experiments to evaluate efficacy and safety in physiological contexts. Comprehensive pharmacological profiling and optimization are essential to maximize specificity and therapeutic potential. Ultimately, integrating rigorous experimental validation with computational findings will be helpful into developing effective KDM4C-targeted anticancer therapies.

## Acknowledgments

MK thanks AIIMS, New Delhi for providing SRship, PKS thanks to Department of Biotechnology, Government of India (BT/RLF/Re-entry/28/2022) for fellowship and infrastructural, institutional support was provided by the Centre for Bio separation Technology (CBST), Vellore Institute of Technology (VIT), India.

## Author contributions

**Conceptualization:** Mukesh Kumar, Anusha P, Soumyadip Mukhopadhyay, Subarnarekha Chowdhury, Prakash K. Shukla.

**Data curation:** Mukesh Kumar, Anusha P, Manoj Phalak, Uma Devi, Prakash K. Shukla.

**Formal analysis:** Mukesh Kumar, Anusha P, Soumyadip Mukhopadhyay, Subarnarekha Chowdhury, Manoj Phalak, Uma Devi, Prakash K. Shukla.

**Funding acquisition:** Prakash K. Shukla.

**Investigation:** Mukesh Kumar, Anusha P, Soumyadip Mukhopadhyay, Subarnarekha Chowdhury, Uma Devi, Prakash K. Shukla.

**Methodology:** Mukesh Kumar, Anusha P, Uma Devi, Prakash K. Shukla.

**Project administration:** Prakash K. Shukla.

**Resources:** Mukesh Kumar, Manoj Phalak, Prakash K. Shukla.

**Software:** Mukesh Kumar, Prakash K. Shukla.

**Supervision:** Manoj Phalak, Prakash K. Shukla.

**Validation:** Mukesh Kumar, Anusha P, Subarnarekha Chowdhury, Manoj Phalak, Uma Devi, Prakash K. Shukla.

**Visualization:** Mukesh Kumar, Anusha P, Subarnarekha Chowdhury, Manoj Phalak, Uma Devi, Prakash K. Shukla.

**Writing – original draft:** Mukesh Kumar, Anusha P, Soumyadip Mukhopadhyay, Subarnarekha Chowdhury, Manoj Phalak, Uma Devi, Prakash K. Shukla.

**Writing – review & editing:** Mukesh Kumar, Prakash K. Shukla.

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
