## [Decision Letter · Decision Letter 0]

20 Jun 2025

PONE-D-25-25228Targeting epigenetic regulators: In-silico discovery of natural inhibitors against histone demethylase KDM4CPLOS ONE

Dear Dr. shukla,

Thank you for submitting your manuscript to PLOS ONE. After careful consideration, we feel that it has merit but does not fully meet PLOS ONE’s publication criteria as it currently stands. Therefore, we invite you to submit a revised version of the manuscript that addresses the points raised during the review process.

Overall, this is a solid and methodologically thorough computational study that identifies natural inhibitors of KDM4C with potential anticancer activity. Although the findings show promise, the manuscript could benefit from a stronger emphasis on the limitations of computational predictions and a more critical discussion. I recommend minor revisions to improve clarity, scientific rigor, and presentation

We look forward to receiving your revised manuscript.

Kind regards,

Akingbolabo Daniel Ogunlakin, Phd

Academic Editor

PLOS ONE

Journal Requirements:

Department of Biotechnology, Government of India (BT/RLF/Re-entry/28/2022).

MK thanks AIIMS, New Delhi for providing SRship, PKS thanks to Department of Biotechnology, Government of India (BT/RLF/Re-entry/28/2022).

Department of Biotechnology, Government of India (BT/RLF/Re-entry/28/2022).

7. When completing the data availability statement of the submission form, you indicated that you will make your data available on acceptance. We strongly recommend all authors decide on a data sharing plan before acceptance, as the process can be lengthy and hold up publication timelines. Please note that, though access restrictions are acceptable now, your entire data will need to be made freely accessible if your manuscript is accepted for publication. This policy applies to all data except where public deposition would breach compliance with the protocol approved by your research ethics board. If you are unable to adhere to our open data policy, please kindly revise your statement to explain your reasoning and we will seek the editor's input on an exemption. Please be assured that, once you have provided your new statement, the assessment of your exemption will not hold up the peer review process.

8. Please include a copy of Table 1, and 3 which you refer to in your text on page 9, and 14.

Reviewers' comments:

Reviewer's Responses to Questions

**Comments to the Author**

1. Is the manuscript technically sound, and do the data support the conclusions?

Reviewer #1: Yes

Reviewer #2: Yes

2. Has the statistical analysis been performed appropriately and rigorously?

Reviewer #1: Yes

Reviewer #2: Yes

3. Have the authors made all data underlying the findings in their manuscript fully available?

Reviewer #1: Yes

Reviewer #2: Yes

4. Is the manuscript presented in an intelligible fashion and written in standard English?

Reviewer #1: Yes

Reviewer #2: Yes

5. Review Comments to the Author

Reviewer #1: Overall, this is a solid and methodologically thorough computational study that identifies natural inhibitors of KDM4C with potential anticancer activity. Although the findings show promise, the manuscript could benefit from a stronger emphasis on the limitations of computational predictions and a more critical discussion. I recommend minor revisions to improve clarity, scientific rigor, and presentation.

Reviewer #2: This study presents a structure-based drug discovery approach to identify natural polyphenolic inhibitors of the histone demethylase KDM4C, a promising epigenetic target implicated in multiple cancers. The use of high-throughput virtual screening, molecular docking, molecular dynamics (MD) simulations, and MM-GBSA free energy calculations to identify pectolinarin and compound 202 as lead candidates is methodologically sound and well-executed. The findings, particularly the superior docking scores of these compounds compared to the reference ligand (6x9, PDBID: 5KR7), stable MD simulations, and favorable ADMET profiles, provide a strong foundation for their potential as KDM4C inhibitors. The study’s focus on natural compounds and its integration of computational tools to advance epigenetic therapeutics is timely and relevant. Authors should take address the comments below before the study can be accepted for publication

Keywords should not repeat words in the title of the manuscript

Authors should propose specific assays (e.g., enzymatic inhibition, cell-based assays, or histone methylation profiling) to guide future studies. This would demonstrate a clear path toward clinical translation.

6. PLOS authors have the option to publish the peer review history of their article (what does this mean?). If published, this will include your full peer review and any attached files.

Reviewer #1: No

Reviewer #2: No

---

## [Author Response · Author response to Decision Letter 1]

2 Aug 2025

Reviewer Comment and answers

The paper titled “Targeting epigenetic regulators: In-silico discovery of natural inhibitors against histone demethylase KDM4C” looks at how to use computer-based methods to find natural compounds that could block KDM4C, an epigenetic regulator linked to several cancers. The study integrates virtual screening, molecular docking, molecular dynamics simulations, and MM-GBSA free energy calculations to evaluate compound binding and stability. The research focuses on a current and significant topic in cancer treatment by suggesting that computer-identified natural compounds, which have good ADMET profiles, could be used as potential epigenetic therapies. Overall, is a relevant and detailed study with translational potential, though some issues related to presentation, clarity, and critical analysis need attention.

We thank the reviewer for the valuable suggestions and the overall positive assessment of our work. We thank the reviewer for valuable suggestions. We have modified the manuscript and incorporated suggestion into the manuscript. The answers to the questions are provided below.

The issues are:

1. Fix the typographical inconsistencies (e.g., “pectolaniarin” vs. “pectolinarin”).

Thank you for pointing out the typographical inconsistency. We have carefully reviewed the manuscript and corrected all instances of the misspelling to ensure uniformity and accuracy throughout the text.

2. The work heavily relies on proprietary software (Schrödinger suite), and it does not discuss reproducibility using open-source tools. Consider including brief commentary on reproducibility or comparison with open-source methods such as AutoDock Vina.

The present research is primarily based on the proprietary Schrödinger suite, which provides robust and highly accurate computational tools for molecular docking, dynamics, and visualization. Nevertheless, to improve reproducibility and accessibility, it is crucial to recognize the possibility of replicating essential elements of the study using open-source alternatives. Tools like AutoDock Vina, PyRx, and Open Babel can be utilized for molecular docking studies, yielding reasonably accurate and reproducible results. Although these tools may not possess some of the advanced features or automation found in Schrödinger, they are extensively used and validated within the scientific community. We also tested manually docked the top compound with the Autodock Vina and found the values very similar to the obtained values from the Schrodinger but not mentioned in this manuscript.

3. The discussion overemphasizes docking scores without accompanying in vitro or in vivo validation, which limits the impact. Add a clearer discussion of the limitations of in silico predictions and the need for experimental confirmation.

We acknowledge the reviewer’s valuable comment regarding the overreliance on docking scores without experimental validation. Indeed, while molecular docking serves as a critical initial step in virtual screening by predicting the binding affinity and orientation of ligands within the target protein’s active site, it is not sufficient alone to confirm the efficacy or biological relevance of the identified hits. Docking scores are influenced by static structural assumptions and do not account for protein flexibility or solvent effects, which can significantly impact ligand binding in a physiological environment. To address this limitation and strengthen the reliability of our findings, we employed molecular dynamics (MD) simulations to evaluate the stability and behavior of the protein–ligand complexes under dynamic conditions. This allowed us to observe the conformational adaptability and consistent binding interactions of the top ten lead compounds, as well as the control molecule, over time. The results revealed stable binding poses and key interactions that were preserved during the 200-ns simulations, suggesting potential biological relevance. Furthermore, to complement the MD simulations, we performed MM/GBSA binding free energy calculations to estimate the thermodynamic favorability of each ligand–protein complex. These calculations provided quantitative insight into the strength of interactions and helped identify residues with significant energetic contributions through per-residue decomposition analysis. These in silico methods together offer a multi-parametric justification of lead selection beyond docking scores alone.

4. The conclusion largely restates the abstract and lacks critical evaluation of study limitations. Use the conclusion to reflect on the limitations of computational-only studies and propose concrete next steps (e.g., planned wet-lab assays).

In our revised conclusion, we have explicitly addressed the inherent limitations of computational-only studies namely, the reliance on in silico predictions without experimental validation, and the potential divergence between computational results and actual biological outcomes. To advance this research, we propose conducting targeted wet-lab assays, such as enzymatic inhibition and cellular activity tests, to validate the computationally identified candidates. These experimental studies are crucial for confirming the therapeutic efficacy and biological relevance of our findings, and will form the basis of our next research phase.

We trust these revisions offer a more balanced perspective and provide a clear roadmap for future work, in accordance with the reviewer’s recommendation.

5. Maintain uniform formatting, with some journal titles abbreviated and others not. A uniform referencing style is needed.

We did the formatting as recommended.

Overall, this is a solid and methodologically thorough computational study that identifies natural inhibitors of KDM4C with potential anticancer activity. Although the findings show promise, the manuscript could benefit from a stronger emphasis on the limitations of computational predictions and a more critical discussion. I recommend minor revisions to improve clarity, scientific rigor, and presentation.

We implemented the suggestion in revised manuscript.

---

## [Decision Letter · Decision Letter 1]

12 Oct 2025

PONE-D-25-25228R1Targeting epigenetic regulators: In-silico discovery of natural inhibitors against histone demethylase KDM4CPLOS ONE

Dear Dr. shukla,

Thank you for submitting your manuscript to PLOS ONE. After careful consideration, we feel that it has merit but does not fully meet PLOS ONE’s publication criteria as it currently stands. Therefore, we invite you to submit a revised version of the manuscript that addresses the points raised during the review process.

**ACADEMIC EDITOR:**Many thanks to the authors for responding positively to the initial concerns. The revision has improved the quality of the submission. However, some grey areas still exist, as highlighted by the reviewers, and these require the authors’ significant attention through another round of revision..

We look forward to receiving your revised manuscript.

Kind regards,

Yusuf Oloruntoyin Ayipo, Ph.D

Academic Editor

PLOS ONE

Journal Requirements:

Additional Editor Comments:

Many thanks to the authors for responding positively to the initial concerns. The revision has improved the quality of the submission. However, some grey areas still exist, as highlighted by the reviewers, and these require the authors’ significant attention through another round of revision.

Reviewers' comments:

Reviewer's Responses to Questions

**Comments to the Author**

1. If the authors have adequately addressed your comments raised in a previous round of review and you feel that this manuscript is now acceptable for publication, you may indicate that here to bypass the “Comments to the Author” section, enter your conflict of interest statement in the “Confidential to Editor” section, and submit your "Accept" recommendation.

Reviewer #1: All comments have been addressed

Reviewer #3: All comments have been addressed

Reviewer #4: (No Response)

Reviewer #5: All comments have been addressed

2. Is the manuscript technically sound, and do the data support the conclusions?

Reviewer #1: Yes

Reviewer #3: Yes

Reviewer #4: Yes

Reviewer #5: Yes

3. Has the statistical analysis been performed appropriately and rigorously?

Reviewer #1: Yes

Reviewer #3: Yes

Reviewer #4: Yes

Reviewer #5: Yes

4. Have the authors made all data underlying the findings in their manuscript fully available?

Reviewer #1: Yes

Reviewer #3: Yes

Reviewer #4: Yes

Reviewer #5: Yes

5. Is the manuscript presented in an intelligible fashion and written in standard English?

Reviewer #1: Yes

Reviewer #3: Yes

Reviewer #4: Yes

Reviewer #5: Yes

6. Review Comments to the Author

Reviewer #1: (No Response)

Reviewer #3: The revised manuscript has significantly improved in clarity, methodological transparency, and balance. The integration of docking, molecular dynamics, and MM-GBSA analyses is technically sound, and the addition of methodological details enhances reproducibility. Conclusions are now appropriately linked to the data presented, and the claims have been calibrated to reflect the computational scope of the study. Figures and tables are informative and free from manipulation.

To further refine the manuscript, I suggest ensuring that all figures include clear legends with units and consistent notation for key parameters, and that RMSD/RMSF values are briefly summarized numerically in the text. Otherwise, the manuscript meets PLOS ONE’s standards for technical soundness, data transparency, and clarity.

Recommendation: Accept for publication (minor textual refinements suggested).

Reviewer #4: Dear Authors,

I thank you for submiotting your manuscript, ''Targeting epigenetic regulators: In-silico discovery of natural inhibitors against histone demethylase KDM4C'' for review. I have completed a thorough review of your work, and noted that your study addresses an important topic in cancer research, leveraging advanced computational methods to identify potential KDM4C inhibitors from natural compounds. However, in its current state, your manuscript has multiple areas that have affected its overall clarity and scientific precision.

1. Consistency and Clarity Issues in Reporting Your Results:

i. Refer to the RMSD values,

I noticed remarkable inconsistencies between the data presented in Figure 3A and your textual descriptions of RMSD values. Example; The figure shows values fluctuating around '1-1.5 Å' for Pectolinarin while the text statement maintained RMSD values at below '0.5 Å'. Similarly, comparing the figure and textual RMSD values for compound 202 and the control complex, there are also notable discrepancies.

Recommendtion; Reconciled these discrepancies to enhance the manuscript's scientific accuracy.

ii. Refer to Figure 5A,

You had labeled this as showing "H-bond" on its Y-axis, however in the text, you referred to it as RMSD and Rg.

Recommendation: Kindly review and clarify the labeling and referencing mismatch.

iii. Refer to the Results section.

Regarding ADMET evaluation, you stated, "The selected top natural compound adheres to Lipinski’s Rule of Five..."

Comment/Recommendation; It's unclear which specific "top natural compound" you refer to, since multiple candidates were identified. You need to clarify if this applies to a any specific compounds or generally to Pectolinarin and compound 202.

2. Captions or Labels for Figures and Tables

Refer to Figures 6, 7, 8:

It seems you used a generic introductory sentence for each of these figures, "Molecular dynamics analysis of KDM4C interactions with the control complex is presented as follows". However, reviewing these figures, I noticed the figures in addition to the control, actually show interactions for Pectolinarin, compound 202, 707, Orientin, and Oroxin.

Recommendation: This gives a view of a misrepresentation or at best an error in reading of the fndings. You may wish to review this.

3. Typos.

I noticed multiple typographycal errors, where words were either overly spaced or not spaced at all. There was also an inconsistent spelling for "Pectolinarin", sometimes it was spelt as "Pectolaniarin", at other times as "pectolarinin". Please identify all these and effect a correction to enhance clarity.

Reviewer #5: Manuscript is sectioned appropriately and clearly written while employing an appropriate methodology

7. PLOS authors have the option to publish the peer review history of their article (what does this mean?). If published, this will include your full peer review and any attached files.

Reviewer #1: No

Reviewer #3: No

Reviewer #4: **Yes: **Patrick Chinazam Nwosu

Reviewer #5: No

---

## [Author Response · Author response to Decision Letter 2]

30 Oct 2025

Reviewer Comment and answers

I thank you for submitting your manuscript, ''Targeting epigenetic regulators: In-silico discovery of natural inhibitors against histone demethylase KDM4C'' for review. I have completed a thorough review of your work, and noted that your study addresses an important topic in cancer research, leveraging advanced computational methods to identify potential KDM4C inhibitors from natural compounds. However, in its current state, your manuscript has multiple areas that have affected its overall clarity and scientific precision.

We thank the reviewer for the valuable suggestions and the overall positive assessment of our work. We have modified the manuscript and incorporated suggestion into the manuscript. The answers to the questions are provided below.

1. Consistency and Clarity Issues in Reporting Your Results:

i. Refer to the RMSD values,

I noticed remarkable inconsistencies between the data presented in Figure 3A and your textual descriptions of RMSD values. Example; The figure shows values fluctuating around '1-1.5 Å' for Pectolinarin while the text statement maintained RMSD values at below '0.5 Å'. Similarly, comparing the figure and textual RMSD values for compound 202 and the control complex, there are also notable discrepancies.

Recommendation; Reconciled these discrepancies to enhance the manuscript's scientific accuracy.

Answer: We sincerely appreciate the reviewer's insightful comments regarding the figure presentations. We have carefully revised the manuscript to rectify these issues. The descriptions for Figures 3A (protein RMSD) and 3B (ligand RMSD within the KDM4C complex) have been made more precise throughout the text. Additionally, we have removed the duplicated image (previously Figure 4A) and consolidated the data presentation. We apologize for the oversights and thank the reviewer for helping us improve the quality of our work.

ii. Refer to Figure 5A,

You had labeled this as showing "H-bond" on its Y-axis, however in the text; you referred to it as RMSD and Rg.

Recommendation: Kindly review and clarify the labeling and referencing mismatch.

Answer: We thank the reviewer for this insight and have revised the manuscript accordingly.

iii. Refer to the Results section.

Regarding ADMET evaluation, you stated, "The selected top natural compound adheres to Lipinski’s Rule of Five..."

Comment/Recommendation; It's unclear which specific "top natural compound" you refer to, since multiple candidates were identified. You need to clarify if this applies to a any specific compounds or generally to Pectolinarin and compound 202.

Answer: We thank the reviewer for this valuable insight. The suggestion has been incorporated into the revised manuscript.

2. Captions or Labels for Figures and Tables

Refer to Figures 6, 7, 8:

It seems you used a generic introductory sentence for each of these figures, "Molecular dynamics analysis of KDM4C interactions with the control complex is presented as follows". However, reviewing these figures, I noticed the figures in addition to the control, actually show interactions for Pectolinarin, compound 202, 707, Orientin, and Oroxin.

Recommendation: This gives a view of a misrepresentation or at best an error in reading of the findings. You may wish to review this.

Answer: We acknowledge the oversight and have implemented the necessary revision in the updated manuscript.

3. Typos.

I noticed multiple typographycal errors, where words were either overly spaced or not spaced at all. There was also an inconsistent spelling for "Pectolinarin", sometimes it was spelt as "Pectolaniarin", at other times as "pectolarinin". Please identify all these and effect a correction to enhance clarity.

Answer: We appreciate the reviewer's keen attention to detail in identifying the typographical inconsistency. We have performed a comprehensive review and have unified the terminology in the revised manuscript to ensure accuracy.

---

## [Decision Letter · Decision Letter 2]

18 Nov 2025

PONE-D-25-25228R2Targeting epigenetic regulators: In-silico discovery of natural inhibitors against histone demethylase KDM4CPLOS ONE

Dear Dr. shukla,

Thank you for submitting your manuscript to PLOS ONE. After careful consideration, we feel that it has merit but does not fully meet PLOS ONE’s publication criteria as it currently stands. Therefore, we invite you to submit a revised version of the manuscript that addresses the points raised during the review process.

**Thanks to the authors for responding positively to the initial concerns. The revision has improved the quality of the submission. However, some grey areas still exist, as highlighted by the reviewers, and these require the authors’ attention through another round of minor revision.. **

We look forward to receiving your revised manuscript.

Kind regards,

Yusuf Oloruntoyin Ayipo, Ph.D

Academic Editor

PLOS ONE

**Journal Requirements:**

**Additional Editor Comments:**

Thanks to the authors for responding positively to the initial concerns. The revision has improved the quality of the submission. However, some grey areas still exist, as highlighted by the reviewers, and these require the authors’ attention through another round of minor revision.

Reviewers' comments:

Reviewer's Responses to Questions

**Comments to the Author**

1. If the authors have adequately addressed your comments raised in a previous round of review and you feel that this manuscript is now acceptable for publication, you may indicate that here to bypass the “Comments to the Author” section, enter your conflict of interest statement in the “Confidential to Editor” section, and submit your "Accept" recommendation.

Reviewer #3: (No Response)

Reviewer #4: All comments have been addressed

2. Is the manuscript technically sound, and do the data support the conclusions?

Reviewer #3: Yes

Reviewer #4: Yes

3. Has the statistical analysis been performed appropriately and rigorously?

Reviewer #3: Yes

Reviewer #4: Yes

4. Have the authors made all data underlying the findings in their manuscript fully available?

Reviewer #3: Yes

Reviewer #4: Yes

5. Is the manuscript presented in an intelligible fashion and written in standard English?

Reviewer #3: Yes

Reviewer #4: Yes

6. Review Comments to the Author

**Reviewer #3: **The manuscript presents a well-structured and technically sound in-silico investigation aimed at identifying natural compounds as potential KDM4C inhibitors. The computational methods including virtual screening, molecular docking, MD simulations, and MM-GBSA analysis are appropriate and rigorously applied. The results are clearly presented and support the conclusions drawn.

To strengthen the manuscript, I recommend the following revisions:

(1) Please simplify or break down some long paragraphs to improve readability, example, The MD simulation revealed stable ligand–protein interaction, and the RMSD remained within acceptable limits throughout the 100 ns simulation, indicating complex stability. Furthermore, hydrogen bonding analysis suggested persistent interactions, confirming the ligand’s potential as an inhibitor, and the MM-GBSA values were favorable, demonstrating strong binding affinity.”

this can be more specific and broken down to this;

The MD simulation revealed stable ligand–protein interactions. The RMSD remained within acceptable limits throughout the 100 ns simulation, indicating overall stability of the complex. Additionally, hydrogen bonding analysis showed persistent interactions. Together with favorable MM-GBSA values, these results suggest strong binding affinity.

(2) Clarify certain methodological details such as ligand library sourcing and the rationale for specific simulation parameters to improve reproducibility;

(3) Where claims appear strong (“inhibits,” “therapeutically effective”), rephrase to more cautious language appropriate for computational predictions;

(4) Include a short limitations paragraph noting the need for biochemical and cellular validation.

These improvements will enhance the manuscript’s clarity and scientific rigor. Overall, the study is well-executed and suitable for publication after minor revision.

**Reviewer #4: **Thank you for submitting your revised manuscript.

I have gone through your recent submission and do not have any further concerns.

7. PLOS authors have the option to publish the peer review history of their article (what does this mean?). If published, this will include your full peer review and any attached files.

Reviewer #3: No

Reviewer #4: **Yes: **Patrick Chinazam Nwosu

---

## [Author Response · Author response to Decision Letter 3]

10 Dec 2025

Review Comments to the Author

Reviewer #3: The manuscript presents a well-structured and technically sound in-silico investigation aimed at identifying natural compounds as potential KDM4C inhibitors. The computational methods including virtual screening, molecular docking, MD simulations, and MM-GBSA analysis are appropriate and rigorously applied. The results are clearly presented and support the conclusions drawn.

We thank the reviewers for the valuable suggestions and the overall positive assessment of our work. We have modified the manuscript and incorporated suggestion into the manuscript. The answers to the questions are provided below.

To strengthen the manuscript, I recommend the following revisions:

(1) Please simplify or break down some long paragraphs to improve readability, example, The MD simulation revealed stable ligand–protein interaction, and the RMSD remained within acceptable limits throughout the 100 ns simulation, indicating complex stability. Furthermore, hydrogen bonding analysis suggested persistent interactions, confirming the ligand’s potential as an inhibitor, and the MM-GBSA values were favorable, demonstrating strong binding affinity.”

this can be more specific and broken down to this;

The MD simulation revealed stable ligand–protein interactions. The RMSD remained within acceptable limits throughout the 100 ns simulation, indicating overall stability of the complex. Additionally, hydrogen bonding analysis showed persistent interactions. Together with favorable MM-GBSA values, these results suggest strong binding affinity.

Answer: We have simplified the text in the revised manuscript.

(2) Clarify certain methodological details such as ligand library sourcing and the rationale for specific simulation parameters to improve reproducibility;

Answer: We have incorporated the library information in the revised manuscript.

(3) Where claims appear strong (“inhibits,” “therapeutically effective”), rephrase to more cautious language appropriate for computational predictions;

Answer: We have taken the suggestions accordingly.

(4) Include a short limitations paragraph noting the need for biochemical and cellular validation.

These improvements will enhance the manuscript’s clarity and scientific rigor. Overall, the study is well-executed and suitable for publication after minor revision.

Answer: We have included in conclusion section.

---

## [Editor Report · Decision Letter 3]

16 Dec 2025

Targeting epigenetic regulators: In-silico discovery of natural inhibitors against histone demethylase KDM4C

PONE-D-25-25228R3

Dear Dr. shukla,

We’re pleased to inform you that your manuscript has been judged scientifically suitable for publication and will be formally accepted for publication once it meets all outstanding technical requirements.

Kind regards,

Yusuf Oloruntoyin Ayipo, Ph.D

Academic Editor

PLOS One

Additional Editor Comments (optional):

The submission is scientifically sound for publication in this title, and all the concerns raised by the respective reviewers regarding the manuscript quality have been satisfactorily addressed. I hereby recommend the manuscript for publication in the current version.
---

## [Editor Report · Acceptance letter]

PONE-D-25-25228R3

PLOS One

Dear Dr. shukla,

I'm pleased to inform you that your manuscript has been deemed suitable for publication in PLOS One. Congratulations! Your manuscript is now being handed over to our production team.

Kind regards,

on behalf of

Dr. Yusuf Oloruntoyin Ayipo

Academic Editor

PLOS One